# The Antimicrobial Resistance (AMR) Rates of *Enterobacterales* in a Rural Hospital from the Eastern Region, Ghana: A Retrospective Study, 2022

**DOI:** 10.3390/antibiotics12081321

**Published:** 2023-08-16

**Authors:** Laura Seijas-Pereda, Carlos Rescalvo-Casas, Marcos Hernando-Gozalo, Vida Angmorkie-Eshun, Eunice Agyei, Vivian Adu-Gyamfi, Isaac Sarsah, Maite Alfonso-Romero, Juan Cuadros-González, Juan Soliveri-de Carranza, Ramón Pérez-Tanoira

**Affiliations:** 1Departamento de Biomedicina y Biotecnología, Facultad de Medicina, Universidad de Alcalá, 28805 Madrid, Spain; carlos.rescalvo@uah.es (C.R.-C.); juan.cuadros@uah.es (J.C.-G.); juan.soliveri@uah.es (J.S.-d.C.); 2Departamento de Microbiología Clínica, Hospital Universitario Príncipe de Asturias, 28805 Madrid, Spain; m.hernando@uah.es; 3Departamento de Química Orgánica y Química Inorgánica, Facultad de Farmacia, Universidad de Alcalá, 28805 Madrid, Spain; 4Laboratory of Microbiology, Saint Dominic’s Hospital, Akwatia P.O. Box 59, Ghana; angmorkieeshun@gmail.com (V.A.-E.); e.agyei@yahoo.es (E.A.); vivian.adu.gy@yahoo.es (V.A.-G.); isaacasare20@yahoo.es (I.S.); alfonsomaite11@yahoo.com (M.A.-R.)

**Keywords:** low- and middle-income countries, antimicrobial resistance, *Enterobacterales*, multidrug-resistance (MDR), Extensively Drug-Resistant (XDR), Ghana

## Abstract

Low- and middle-income countries bear a disproportionate burden of antimicrobial resistance and often lack adequate surveillance due to a paucity of microbiological studies. In this 2022 study, our goal was to contribute to a more precise antimicrobial treatment by understanding the prevalence of resistance in a rural environment, promoting antibiotic stewardship, and raising awareness about antimicrobial resistance. We assessed the prevalence of Multidrug-Resistant (MDR) and Extensively Drug-Resistant (XDR) *Enterobacterales* in clinical samples from 2905 patients being treated at Saint Dominic’s Hospital, Akwatia, in the countryside of the Eastern Region, Ghana, in the year 2022. To this purpose, the samples were cultured on agar plates prepared in the laboratory using purified Oxoid™ Thermo Scientific™ agar (Thermo Fisher Scientific; Waltham, MA, USA). Cystine Lactose Electrolyte-Deficient (CLED) agar was used for urine samples, while blood agar, chocolate agar, and MacConkey agar were used for the rest of the specimens tested (HVS, blood, BFA, sputum). Antimicrobial susceptibility was determined on site using the disc diffusion method (Kirby-Bauer test). MDR bacteria accounted for more than half (53.7%) of all microorganisms tested for three or more antibiotics and 37.3% of these were XDR. Multivariate regression analysis was performed to identify risk factors associated with acquiring MDR/XDR bacteria. The results showed an increased likelihood of MDR acquisition linked to being male (OR 2.39, *p* < 0.001 for MDR and OR 1.95, *p* = 0.027 for XDR), higher age (OR 1.01, *p* = 0.049 for MDR), non-sputum samples (OR 0.32, *p* = 0.009 for MDR), and urine samples (OR 7.46, *p* < 0.001 for XDR). These findings emphasize the urgency for surveillance and control of antimicrobial resistance; to this end, making accurate diagnostics, studying the microorganism in question, and conducting susceptibility testing is of the utmost importance.

## 1. Introduction

Antimicrobial resistance (AMR) is a well-known health problem described in a wide variety of microorganisms. The 2014 World Health Organization (WHO) report on global resistance [1] emphasized the significance of resistance in common bacteria such as *Escherichia coli* and *Klebsiella pneumoniae,* since the lack of homogeneous surveillance is astonishingly widespread [2].

AMR is a complex and multifactorial issue, particularly prevalent in low- and middle-income countries (LMIC) such as Ghana. These countries have all the favorable conditions for its development and spread, including overstretched public health systems, insufficient access to diagnostics, overcrowding, inadequate access to safe drinking water and sanitation, and a lack of regulations for antibiotic use [3,4]. In rural Africa, a high percentage of animals intended for human consumption have been found to be colonized by multi-resistant bacteria, and farmers generally lack knowledge of appropriate husbandry systems and antibiotic use. Consequently, there is an increasing concern about the emergence and transmission of antimicrobial resistance to humans via the food supply chain and environmental factors [5]. In addition, close interaction and cohabitation with animals is common in rural Africa, which facilitates the spread of bacterial infections and resistance [3,6,7]. 

The WHO published the global list of priority pathogens in 2017 [8], in which *Enterobacterales* appear among the highest critical category, due to their development of AMR. The *Enterobacterales* family is broad and includes many genera of clinical interest, within which carbapenemases and carriage of extended-spectrum β-lactamases (ESBLs) are the most significant resistance mechanisms. Furthermore, resistance to fluoroquinolones is present in at least half of the clinical isolates of *Escherichia coli* reported in many parts of the world [8,9,10,11,12]. All this makes these infections very difficult to treat, especially since carbapenems such as imipenem, meropenem, and ertapenem have been used as a last resort in the treatment of patients infected with *Enterobacteriaceae* [13,14]. Their high prevalence and association with mortality and morbidity, particularly *E. coli* and *K. pneumoniae*, which are the main agents involved in severe sepsis and septic shock, make it essential to study AMR in *Enterobacterales* and to choose the appropriate antibiotic treatment [12,15,16,17]. 

The problem of antimicrobial resistance is accentuated in rural hospitals in LMIC due to the limited availability of antibiotics and the uncontrolled overuse of broad-spectrum antibiotics, but also due to the lack of etiological diagnosis and the absence of antimicrobial susceptibility studies [12,18]. An accurate microbiological diagnosis of infectious diseases and the administration of an appropriate treatment are hindered by the lack of material resources and laboratory personnel trained in these techniques [8,19].

At this point, it must be said that several studies have reported a higher-than-usual rate of antibiotic prescriptions in the area and relevant research points out its correlation with antimicrobial susceptibility in Ghana, both in inpatient and outpatient settings [12,20,21,22,23,24]. Hence, the main aim of our study shall be to describe the etiology and antimicrobial susceptibility of *Enterobacterales* strains isolated from patients treated for various types of infections at a hospital located in the countryside of the Eastern Region of Ghana, an area where these kinds of studies are scarce. By addressing such AMR challenges in this rural healthcare environment, we aim to contribute to a more precise antimicrobial treatment based on the prevalence of resistance shown in the aforementioned setting, thus improving antibiotic stewardship and promoting antimicrobial resistance awareness.

## 2. Results

### 2.1. Population

A total of 2905 patients were included in the study. Table 1 shows that there were 1920 (66.1%) females and 985 males (33.9%), with a median age of 30 ± 28 years. The most requested analysis were urine samples with 53.3%, followed by 19.6% high vaginal swabs (HVS), 15.1% blood cultures, 7% body fluids and aspirates (BFA), and 5.1% of sputum requests. 

Urine samples were especially requested for females, also showing a higher percentage of positivity than males. As for the rest of the sample types (except HVS), both men and women were equally represented and showed similar positivity, except in BFA, where women had more positivity. Contrary to all other samples, blood cultures were mainly requested in infants.

Leucocyte percentages were obtained by manual counting from the Gram stain of each sample, when possible, as they serve as indicators of the severity of the infection.

### 2.2. Etiological Diagnosis

At least one microorganism was isolated from growth on the agar plate in 872 (30%) cultures. The main bacteria isolated were members of the *Enterobacterales* family (395; 45.3%) including *E. coli* (n = 144; 16.5%), *Klebsiella* spp. (n = 138; 15.8%), non-identified *Enterobacterales* (79; 9.1%), *Proteus* spp. (23; 2.6%), and *Enterobacter* spp. (11; 1.3%). Other microorganisms isolated were *Candida* spp. (n = 232; 26.6%), coagulase negative *Staphylococcus* (n = 174; 20.0%), *Pseudomonas* spp. (27; 3.1%), *Streptococcus* spp. (22; 2.5%), and *Staphylococcus aureus* (17; 1.9%).

*Enterobacterales* were mainly isolated from urine samples (n = 268; 67.8%) followed by HVS (n = 74; 18.7%), sputum (23; 5.8%), BFA (n = 21; 5.3%), and blood (n = 9; 2.3%). Table 2, Table 3, Table 4, Table 5, Table 6 and Table 7 present the main *Enterobacterales* isolated from each sample type. It is important to note that, as explained in Section 4, we categorized some isolates as ‘Non-identified *Enterobacterales’* because we could not further identify their genus and species.

### 2.3. Sensitivity Analysis of Enterobacterales

The most prevalent *Enterobacterales* were analyzed for the available antibiotics (Table 2, Table 3, Table 4, Table 5, Table 6 and Table 7). The most frequently tested drugs were Amikacin, Cefixime, Ceftriaxone, Ciprofloxacin, Gentamicin, and Meropenem.

#### 2.3.1. Urine

From the three hundred and fifty-two *Enterobacterales* isolated from urine samples, the distribution of species was: one hundred and fourteen were *E. coli*, 87 *Klebsiella* spp., ten were *Proteus* spp., seven were *Enterobacter* spp., and fifty were non-identified *Enterobacterales*. For the sensitivity analysis in Table 2, we excluded the non-identified *Enterobacterales*, as they will be represented in Table 7. 

In this section, the most common bacterium found was *E. coli*, which exhibited a higher sensitivity to Amikacin (91.7%) and Nitrofurantoin (69.7%). However, *E. coli* showed higher resistance rates to Pipemedic and Nalidixic acid (95 and 76.4%, respectively) and all Cephalosporins (ranging from 55% to 81%).

The second most frequent microorganism isolated from urine was *Klebsiella* spp., which demonstrated higher sensitivity to Amikacin (95.3%) and Levofloxacin (81.5%), while demonstrating higher resistance to Amoxicillin/clavulanic acid (95.8%), Cephalosporins (57–100%), Pipemedic acid (95.7%), and Tetracycline (73.7%).

**Table 2 antibiotics-12-01321-t002:** Antimicrobial resistance profile obtained by Kirby-Bauer test of different *Enterobacterales* isolated from urine cultures. R (Resistant): a microorganism is classified as resistant when there is a high likelihood of therapeutic failure even with increased exposure.

	*E. coli* (114) R/Total Strains Tested	*Enterobacter* spp. (7) R/Total Strains Tested	*Klebsiella* spp. (87) R/Total Strains Tested	*Proteus* spp. (10) R/Total Strains Tested
Amoxicillin/clavulanic acid	25/26 (96.2%)	3/3 (100%)	23/24 (95.8%)	2/3 (66.7%)
Amikacin	9/108 (8.3%)	0/7 (0%)	4/85 (4.7%)	1/10 (10.0%)
Ampicillin	2/3 (66.7%)	1/1 (100%)	2/2 (100%)	1/1 (100%)
Cefixime	29/48 (60.4%)	1/1 (100%)	17/30 (56.7%)	1/3 (33.3%)
Ceftazidime	33/41 (80.5%)	4/4 (100%)	36/42 (85.7%)	4/5 (80.0%)
Ceftriaxone	57/103 (55.3%)	3/6 (50.0%)	59/84 (70.2%)	4/9 (44.4%)
Cefotaxime	22/32 (68.8%)	1/2 (50.0%)	12/20 (60.0%)	2/3 (66.7%)
Cefuroxime	9/15 (60.0%)	3/3 (100%)	8/8 (100%)	1/2 (50.0%)
Ciprofloxacin	61/109 (56.0%)	5/7 (71.4%)	49/80 (61.3%)	4/10 (40.0%)
Meropenem	2/13 (15.4%)	0/1 (0%)	3/4 (75.0%)	1/2 (50.0%)
Levofloxacin	6/35 (17.1%)	0/4 (0%)	5/27 (18.5%)	0/3 (0%)
Gentamicin	49/104 (47.1%)	4/7 (57.1%)	41/75 (54.7%)	3/9 (33.3%)
Tetracycline	13/21 (61.9%)	2/2 (100%)	14/19 (73.7%)	2/3 (66.7%)
Nitrofurantoin	27/89 (30.3%)	4/4 (100%)	37/70 (52.9%)	4/8 (50.0%)
Norfloxacin	7/17 (41.2%)	-	7/11 (63.6%)	0/1 (0%)
Trimethoprim/sulfamethoxazole	1/1 (100%)	1/1 (100%)	2/2 (100%)	1/1 (100%)
Nalidixic acid	68/89 (76.4%)	5/6(83.3%)	42/68 (61.8%)	4/5 (80.0%)
Pipemedic acid	19/20 (95.0%)	4/4 (100%)	22/23 (95.7%)	3/3 (100%)

#### 2.3.2. High Vaginal Swab

In this section of HVS, we encountered a total of 74 isolates, explained in Table 3, except for one *Enterobacter* spp. isolate that was included in the Non-identified *Enterobacterales.* We added this *Enterobacter* spp. to the non-identified group for the sensitivity analysis because an individual isolate holds no statistical significance. 

Among the isolates obtained from high vaginal swabs, the most frequently identified was *Klebsiella* spp. Notably, *Klebsiella* spp. demonstrated the highest sensitivity to Amikacin (96.2%), Meropenem (92.9%), and Levofloxacin (100%), while the highest resistance rates were to Amoxicillin/clavulanic acid (100%) and all Cephalosporins (63–100%). 

The second most common isolate found was *E. coli*, with the highest sensitivity rates to Amikacin (95.8%) and Meropenem (62.5%). Conversely, the highest resistances to Amoxicillin/clavulanic acid (95.8%), Cephalosporins (58–100%), and Ampicillin (100%).

**Table 3 antibiotics-12-01321-t003:** Antimicrobial resistance profile obtained by Kirby-Bauer test of different *Enterobacterales* isolated from High Vaginal Swabs cultures. R (Resistant): a microorganism is classified as resistant when there is a high likelihood of therapeutic failure even with increased exposure.

	*E. coli* (24) R	*Klebsiella* spp. (27) R	*Proteus* spp. (6) R	Non-Identified *Enterobacterium* (17) R
Amoxicillin/clavulanic acid	3/4 (75.0%)	4/4 (100%)	-	1/1 (100%)
Amikacin	1/24 (4.2%)	1/26 (3.8%)	1/6 (16.7%)	1/13 (7.7%)
Ampicillin	4/4 (100%)	1/1 (100%)	0/1 (0%)	3/5 (60.0%)
Chloramphenicol	1/2 (50.0%)	-	-	0/3 (0%)
Cefixime	6/9 (66.7%)	8/10 (80.0%)	0/2 (0%)	4/4 (100%)
Ceftazidime	3/4 (75.0%)	7/9 (77.8%)	-	2/2 (100%)
Ceftriaxone	14/24 (58.3%)	15/24 (62.5%)	0/6 (0%)	9/15 (60.0%)
Cefotaxime	8/9 (88.9%)	6/8 (75.0%)	0/2 (0%)	8/11 (72.7%)
Cefuroxime	6/6 (100%)	4/4 (100%)	0/1 (0%)	6/8 (75.0%)
Ciprofloxacin	10/22 (45.5%)	11/27 (40.7%)	2/6 (33.3%)	5/16 (31.3%)
Meropenem	6/16 (37.5%)	1/14 (7.1%)	0/3 (0%)	7/12 (58.3%)
Levofloxacin	1/3 (33.3%)	0/4 (0%)		0/1 (0%)
Gentamicin	12/23 (52.2%)	9/26 (34.6%)	2/6 (33.3%)	3/14 (21.4%)
Tetracycline	3/5 (60.0%)	2/3 (66.7%)	-	2/5 (40.0%)
Nitrofurantoin	0/1 (0%)	0/1 (0%)	-	-
Norfloxacin	0/2 (0%)	-	-	-
Trimethoprim/sulfamethoxazole	3/3 (100%)	1/1 (100%)	0/1 (0%)	3/6 (50.0%)

#### 2.3.3. Blood

A small number of *Enterobacterales* were isolated from blood cultures, totaling nine isolates represented in Table 4. Among them, six were non-identified *Enterobacterales*, two *Klebsiella* spp., and one *Escherichia coli*. Given their limited quantity, we grouped them together for sensitivity analysis.

The results indicated that this group showed high sensitivity to Amikacin (100%) and Ciprofloxacin (80%), while Cefixime was the antibiotic for which the highest rate of resistance was found, with 83.3% of the isolates resistant to this antibiotic. 

**Table 4 antibiotics-12-01321-t004:** Antimicrobial resistance profile obtained using the Kirby-Bauer test of different *Enterobacterales* isolated from blood cultures. This group includes six non-identified *Enterobacterium*, two *Klebsiella* spp., and one *Escherichia coli*. R (Resistant): a microorganism is classified as resistant when there is a high likelihood of therapeutic failure even with increased exposure.

	*Enterobacterium* (9) R
Amikacin	0/9 (0%)
Ceftriaxone	4/8 (50.0%)
Ciprofloxacin	1/5 (20.0%)
Cefotaxime	2/7 (28.6%)
Cefixime	5/6 (83.3%)
Gentamicin	3/8 (37.5%)
Meropenem	2/6 (33.3%)

#### 2.3.4. Body Fluids and Aspirates

A small number of *Enterobacterales* were isolated from BFA, totaling twenty-one isolates including two *Escherichia coli*, eleven *Klebsiella* spp., five *Proteus* spp., and three non-identified *Enterobacterium*. Given their limited quantity, we grouped them together for sensitivity analysis as shown in Table 5.

As a group, the *Enterobacterales* were more sensitive to Amikacin and Meropenem, with both antibiotics showing over 90% of sensitivity among the isolates. On the other hand, the group exhibited higher rates of resistance to Cefuroxime (83.3%) and Tetracycline (100%). 

**Table 5 antibiotics-12-01321-t005:** Antimicrobial resistance profile obtained by Kirby-Bauer test of different *Enterobacterales* isolated from body aspirates and fluids cultures. This group includes two *Escherichiacoli*, eleven *Klebsiella* spp., five *Proteus* spp., and three non-identified *Enterobacterium*. R (Resistant): a microorganism is classified as resistant when there is a high likelihood of therapeutic failure even with increased exposure.

	*Enterobacterium* (21) R
Amoxicillin/clavulanic acid	3/4 (75.0%)
Amikacin	1/20 (5.0%)
Ampicillin	3/4 (75.0%)
Ceftazidime	3/5 (60.0%)
Ceftriaxone	6/20 (30.0%)
Ciprofloxacin	3/19 (15.8%)
Cefotaxime	4/10 (40.0%)
Cefuroxime	5/6 (83.3%)
Cefixime	3/8 (37.5%)
Gentamicin	4/19 (21.1%)
Meropenem	1/10 (10.0%)
Tetracycline	4/4 (100%)

#### 2.3.5. Sputum

A small number of *Enterobacterales* were isolated from sputum, a total of 23. Among them, three were *Escherichia coli*, three *Enterobacter* spp., eleven *Klebsiella* spp., two *Proteus* spp., and four non-identified *Enterobacterium*. Given their limited quantity, we grouped them together for sensitivity analysis showed in Table 6. *Enterobacterales* as a group were more sensitive to Amikacin, with 100% of isolates, and Ciprofloxacin (87%), while higher resistance rates were seen in Azithromycin (66,7%) and Cefotaxime, with up to 80% of isolates being resistant to this antibiotic. 

**Table 6 antibiotics-12-01321-t006:** Antimicrobial resistance profile obtained by Kirby-Bauer test of different *Enterobacterales* isolated from Sputum cultures. This group includes three *Escherichia coli*, three *Enterobacter* spp., eleven *Klebsiella* spp., two *Proteus* spp., and four non-identified *Enterobacterales*. R (Resistant): a microorganism is classified as resistant when there is a high likelihood of therapeutic failure even with increased exposure.

	*Enterobacterium* (23) R
Amikacin	0/22 (0%)
Azithromycin	10/15 (66.7%)
Ceftazidime	3/4 (75.0%)
Ceftriaxone	6/21 (28.6%)
Ciprofloxacin	3/23 (13.0%)
Cefotaxime	4/5 (80.0%)
Cefixime	7/12 (58.3%)
Gentamicin	4/20 (20.0%)
Meropenem	1/12 (8.3%)

#### 2.3.6. Enterobacterales

In this section, we considered the 395 *Enterobacterales* from our data, without separating them for sample type (Table 7). We found the most common to be *E. coli* (36.5%), closely followed by *Klebsiella* spp. (34.9%). Among all microorganisms studied, higher resistance rates were observed for Ampicillin, Cefalosporines, Tetracycline, Nalidixic, and Pipemedic Acid. On the other hand, increased sensitivity was found for Amikacin, Meropenem, and Levofloxacin.

**Table 7 antibiotics-12-01321-t007:** Antimicrobial resistance profile obtained by Kirby-Bauer test of the total of each *Enterobacterales* isolated from the total cultures. R (Resistant): a microorganism is classified as resistant when there is a high likelihood of therapeutic failure even with increased exposure.

	*E. coli* (144) R	*Enterobacter* spp. (11) R	*Klebsiella* spp. (138) R	*Proteus* spp. (23) R	Non-Identified *Enterobacterium* (79) R
Amoxicillin/Clavulanic acid	28/31 (90.3%)	4/4 (100%)	29/31 (93.6%)	3/4 (75.0%)	10/10 (100%)
Amikacin	10/137 (7.3%)	0/11 (0%)	5/136 (3.7%)	3/23 (13.0%)	3/74 (4.1%)
Ampicillin	8/9 (88.9%)	1/1 (100%)	6/6 (100%)	1/3 (33.3%)	3/5 (60.0%)
Azithromycin	0/1 (0%)	2/2 (100%)	5/7 (71.4%)	1/1 (100%)	3/6 (50.0%)
Chloramphenicol	2/5 (40.0%)	0/1 (0%)	-	1/1 (100%)	0/3 (0%)
Cefixime	37/59 (62.7%)	2/3 (66.7%)	29/51 (56.9%)	2/7 (28.6%)	19/28 (67.9%)
Ceftazidime	36/46 (78.2%)	5/5 (100%)	45/55 (81.8%)	6/7 (85.7%)	20/24 (83.3%)
Ceftriaxone	75/132 (56.8%)	3/10 (30.0%)	83/132 (62.9%)	5/21 (23.8%)	39/75 (52.0%)
Cefotaxime	34/45 (75.6%)	1/3 (33.3%)	24/38 (63.2%)	2/7 (28.6%)	11/21 (52.4%)
Cefuroxime	17/23 (73.9%)	3/3 (100%)	16/17 (94.1%)	2/5 (40.0%)	9/14 (64.3%)
Ciprofloxacin	75/136 (55.1%)	5/11 (45.5%)	63/131 (48.1%)	6/22 (27.3%)	39/72 (54.2%)
Meropenem	10/32 (31.3%)	0/3 (0%)	5/31 (16.1%)	1/8 (12.5%)	8/23 (34.8%)
Levofloxacin	7/38 (18.4%)	0/5 (0%)	5/33 (15.2%)	1/4 (25.0%)	2/15 (13.3%)
Gentamicin	63/131 (48.1%)	4/11 (36.4%)	54/125 (43.2%)	7/21 (33.3%)	30/73 (41.1%)
Tetracycline	18/28 (64.3%)	3/3 (100%)	18/24 (75.0%)	3/4 (75.0%)	10/14 (71.4%)
Nitrofurantoin	27/90 (30.0%)	4/4 (100%)	38/72 (52.8%)	4/8 (50.0%)	19/40 (47.5%)
Norfloxacin	7/19 (36.8%)	-	7/12 (58.3%)	0/1 (0%)	3/5 (60.0%)
Trimethoprim/sulfamethoxazole	6/6 (100%)	1/1 (100%)	4/4 (100%)	1/3 (33.3%)	3/6 (50.0%)
Nalidixic acid	68/89 (76.4%)	5/6 (83.3%)	42/69 (60.9%)	4/5 (80.0%)	31/37 (83.8%)
Pipemedic acid	19/20 (95.0%)	4/4 (100%)	23/24 (95.8%)	3/3 (100%)	7/7 (100%)

In Table 8, we present resistance rates for the different *Enterobacterales* identified in our study, without considering the specimen they originated from. We observed a high prevalence of resistances among the main microorganisms, not only concerning specific groups of antibiotics but also Multidrug-Resistant (MDR) strains, which accounted for nearly half of all microorganisms tested. Additionally, Extensively Drug-Resistant (XDR) bacteria were also highly prevalent, although less common compared to MDR strains.

Univariate and multivariate analyses included age, sex, leucocytes from the Gram stain, and sample type. In the univariate analysis, we identified significant variables that showed associations with MDR/XDR bacteria that were later included in the multivariate analysis.

The univariate analysis, presented in Table 9A, reveals significant relationships between having MDR/XDR bacteria and being male, as well as with older age. Additionally, urine samples show a significantly higher occurrence of MDR/XDR bacteria compared to other sample types, while sputum samples exhibit fewer MDR bacteria, but not XDR bacteria.

Considering XDR bacteria (Table 9B), we found that higher age, being male, and urine samples are still associated with this occurrence. However, when conducting the multivariate analysis, only being male and having urine samples are significant risk factors.

**Table 9 antibiotics-12-01321-t009:** (A and B). Risk factors associated with having Multidrug-Resistant (MDR) (A) and Extensive Drug-Resistant (XDR) (B) bacteria. nrepresents the number of patients from which we have data for each MDR/XDR bacteria group. * No cases of XDR bacteria found in this group.

A. MDR	Univariate OR (95% CI)	*p*-Value	Multivariate OR (95% CI)	*p*-Value
Age (n = 420)	1.01 (1.01–1.02)	0.002	1.01 (1.00–1.02)	0.049
Male (n = 142)	2.35 (1.52–3.61)	<0.001	2.39 (1.49–3.83)	<0.001
Leucocytes (n = 339)	1.01 (1.00–1.02)	0.279		
Sample Type				
Urine (n = 270)	1.56 (1.05–2.32)	0.028	Not included *p* > 0.05	
High Vaginal Swabs (n = 75)	0.75 (0.46–1.24)	0.263		
Blood (n = 25)	1.95 (0.80–4.76)	0.145		
Body fluids and aspirates (n = 25)	0.46 (0.20–1.06)	0.069		
Sputum (n = 34)	0.48 (0.24–0.98)	0.044	0.33(0.14–0.76)	0.009
B. XDR	Univariate OR (95% CI)	*p*-Value	Multivariate OR (95% CI)	*p*-Value
Age (n = 420)	1.02 (1.01–1.03)	0.004	Not included *p* > 0.05	
Male (n = 142)	1.88 (1.14–3.08)	0.013	1.95 (1.08–3.54)	0.027
Leucocytes (n = 339)	1.01 (1.00–1.02)	0.223		
Sample Type				
Urine (n = 270)	5.21 (2.60–10.46)	<0.001	7.46 (2.55–21.87)	<0.001
High Vaginal Swabs (n = 75)	0.33 (0.14–0.79)	0.013	Not included *p* > 0.05	n.s.
Blood (n = 25)	0.36 (0.08–1.57)	0.176		
Body fluids and aspirates (n = 25)	0.36 (0.08–1.57)	0.176		
Sputum * (n = 34)	-	-		

## 3. Discussion

This study was conducted to determine the drug susceptibility profile of bacterial strains isolated from different types of samples analyzed at a hospital in the Eastern Region of Ghana. Bacterial growth was detected in 30% of the 2905 samples tested. The most frequent patient profile whose specimen was analyzed in the microbiology laboratory was a 30-year-old female with a presumed urinary tract infection [25]. The median age of infection in Africa is lower than that of other corners of the world, such as Europe, due to differences in the population pyramid and living conditions [26]. 

The *Enterobacteriaceae* family was the most frequently isolated microorganisms, accounting for 45.3% of the isolates, with a predominance of *E. coli* (16.5%) and *Klebsiella* spp. (15.8%), which is consistent with findings from several other studies [9,12,27,28]. Urine samples were the primary source for the majority of *Enterobacterales* isolates (53.3%), reflecting a common trend in resource-poor areas such as Ghana, where the availability of other sample types for study is limited, thus restricting the options clinicians can request [29]. Due to the scarcity of data on *Enterobacterales* in Africa, making a comparison proves rather challenging, yet our results show striking similarities with findings from other regions, such as Sierra Leone [1,27].

The culture positivity rate was approximately 30%, except for sputum and blood samples, which exhibited higher rates, reaching up to 50%. Sputum samples, being rich in microorganisms, may show growth that is not always clinically relevant [30]. Interestingly, the median age for blood culture requests was one year. In affluent countries, these types of infection in infants are decreasing due to improved control and prevention techniques [31]; it is important to implement these measures in Ghana and other countries affected by this affliction.

We found high resistance rates in Ghana, consistent with previous studies and WHO reports worldwide. Particularly, our study showed a high prevalence of *Enterobacterales* strains displaying high resistance to Amoxicillin / clavulanic acid and fluoroquinolones, both important oral treatment options. Additionally, we looked into the susceptibility of third- and fourth-generation cephalosporins (i.e., cefotaxime and cefepime), which are used as indicators for extended-spectrum beta-lactamase (ESBL) production when showing resistances to this class of antibiotics. We found the overall prevalence of ESBL-producing strains to be over 60%. Notably, our results indicate higher resistances compared to other studies conducted in low- and middle-income countries [28,32,33,34]. This discrepancy may be attributed to the inferior sanitation and healthcare facilities in rural areas, such as Akwatia, compared to African capitals and larger cities, which have been the subject of study more frequently. For this reason, the findings of this study shed light on the concerning issue of antibiotic resistance in rural areas of LMICs, adding significant value to the existing knowledge on countryside studies and allowing for meaningful comparisons with regions having more comprehensive data.

Among the *Enterobacterales* analyzed, approximately 50% displayed multidrug resistance (MDR), except for *Proteus* spp., with 30%. Of these MDR bacteria, 37.3% were XDR bacteria, alarmingly high for such a severe resistance mechanism. On the other hand, susceptibility to nitrofurantoin, a commonly used oral treatment for urinary tract infections, varied among isolates, with *Klebsiella spp.* exhibiting a low susceptibility rate of 48%, while it was high for *E. coli*, at 70%. It should also be noted that pathogens with intrinsic resistance to nitrofurantoin, such as *Proteus* spp. or other bacteria with the urease enzyme, were infrequent. As a positive aspect, the “in vitro” susceptibility profiles of isolated microorganisms to carbapenems were high, with a rate close to 75%. These results underscore the urgent need for effective antimicrobial stewardship programs and targeted interventions in LMICs’ rural settings to battle the rising threat of antibiotic resistance and preserve the effectiveness of critical antibiotics, such as carbapenems.

Finally, our study demonstrated a significant association between the factors of gender (i.e., being male) and age (i.e., being older) with an increased risk of having an MDR bacterial infection. Moreover, we observed that MDR/XDR bacteria were predominantly isolated from urine samples, which constituted the largest proportion in this study. This underlines the importance of considering the infection site when determining the appropriate treatment approach. Previous similar studies have also reported a link between bacterial infections in urine and factors such as residing in rural areas, being uncircumcised, and experiencing frequent urination, which could be more prevalent in older age groups. Additionally, the frequency of resistance in urinary tract infections is higher due to patients often receiving prescribed antibiotics and self-administering them without supervision, particularly in regions where self-medication practices are common [35,36,37].

Overall, with this study, we aim to contribute to the improvement of antimicrobial treatment accuracy, based on the prevalence of resistance observed in this rural area of Ghana. These findings will serve as a valuable criterion for optimizing antibiotic use and informing future planning strategies for countryside regions, effectively addressing the challenge of antibiotic resistance.

Though it might not be obvious to the naked eye, the main drawback of this study would be, to our mind, its retrospective nature, which may have influenced the statistical analysis due to the availability of certain data being in short supply. As pointed out in the methods section, clinical and epidemiological data were retrospectively collected from patients’ medical records, and in some cases, some information was missing. Another significant limitation of this study is that all laboratory analyses were conducted in Ghana, where resources and the availability of materials are limited. Consequently, we faced numerous difficulties in identifying some bacterial isolates to genus and species level, as well as determining resistance mechanisms.

## 4. Materials and Methods

### 4.1. Location

This retrospective study was conducted by collecting data from all bacterial infections diagnosed at the microbiology laboratory of Saint Dominic Hospital (counting on 320 beds) in Akwatia, Eastern Region of Ghana. This is a rural zone located 124 km from Accra, the capital of the country. The hospital is the medical referral centre for the area, aiding around 80,000 people, and it is surrounded by several smaller healthcare centres. The hospital’s laboratory is well supplied with all basic materials, equipment, trained and professional staff (physicians and technicians), etc.

### 4.2. Inclusion Criteria

This retrospective study included all results of bacteria isolated from cultures at the microbiology laboratory of Saint Dominic during the year 2022. Anonymized data were collected from the hospital records of all patients who met the inclusion criteria.

Samples of interest were urine, blood, body fluids and aspirates (BFA), high vaginal swabs (HVS), and sputum. Analysis was made only after a physician’s request due to an infection suspicion. We removed 7 patients with contaminated cultures from the study, leaving a total of 2905 patients.

### 4.3. Specimen’s Collection and Processing

Patients were requested to attend the laboratory for specimen collection; however, some specimens were delivered by nursing staff due to specific procedures, being from inpatients, or due to the patient’s incapacity to reach the laboratory. The collection procedure was performed under sterile conditions, tailored to the specific body site. All samples were collected and kept in sterile containers appropriate for each sample type.

Initially, samples were analyzed by microscopy using Gram staining. Subsequently, the samples were cultured on agar plates prepared in the laboratory using purified Oxoid™ Thermo Scientific™ agar. Cystine Lactose Electrolyte-Deficient (CLED) agar was used for urine samples, while blood agar, chocolate agar, and MacConkey agar were used for the rest of the specimens tested (HVS, blood, BFA, sputum).

The bacterial isolates were characterized using colony morphology and Gram staining reaction, and through a panel of biochemical tests following the standard microbiological procedure. Gram-positive cocci were distinguished and recognized based on Gram stain, blood agar hemolysis patterns, colonial characteristics, a catalase test, a coagulase test, and a mannitol fermentation test. Gram-negative bacteria were identified based on Gram reaction, colony morphology (visual culture characteristics of a bacterial colony on an agar plate) pigmentation, on triple sugar iron agar (TSI) fermentation of glucose and lactose and H2S production, formation of indole, and citrate utilization, and urea hydrolysis [38].

The combination of these results typically led to the identification of a specific bacterial genus and species, as shown in Appendix A. However, in some cases, certain combinations of results did not match any of the microorganisms within the scope of our technique. In such cases, we could only report the identification as far as we could determine, such as indicated in the results, “Non-identified *Enterobacterium*”.

For antimicrobial susceptibility studies of *Enterobacterales*, we employed the Kirby-Bauer test or disc-diffusion method with a 0.5 McFarland inoculum in 0.9% NaCl saline on Mueller Hinton agar (Oxoid™ Thermo Scientific™). Various antibiotics discs were used, including meropenem, ceftriaxone, cefuroxime, sulfamethoxazole-trimethoprim, gentamicin, cefoxitin, cefixime, clindamycin, cloxacillin, chloramphenicol, erythromycin, nalidixic acid, pipemidic acid, amoxicillin/clavulanic acid, ampicillin, azithromycin, cefotaxime, ceftazidime, levofloxacin, nitrofurantoin, norfloxacin, ciprofloxacin, amikacin, and tetracycline. The choice of antibiotics and testing methods varied depending on the sample type, isolated pathogen, and availability of discs; 90 mm glass plates were used, with a maximum of 6 disks per plate.

Antimicrobial susceptibility for each antibiotic was determined by measuring the inhibition halo according to the Clinical and Laboratory Standards Institute (CLSI) interpretive chart for susceptibility tests 2022 [39], the European Committee on Antimicrobial Susceptibility Testing (EUCAST), and/or the U.S. Food and Drug Administration (FDA) guidelines. A bacterial isolate was considered non-susceptible to an antimicrobial agent if it tested resistant, intermediate, or non-susceptible based on these clinical breakpoints as interpretive criteria.

To define Multidrug-Resistant (MDR) bacteria and Extensively Drug-Resistant (XDR) bacteria, we considered only acquired antimicrobial resistance and not intrinsic resistance (if existing). MDR bacteria were defined as those non-susceptible to at least one agent in three or more antimicrobial categories, while XDR bacteria were those non-susceptible to at least one agent in five or more antimicrobial categories [19].

### 4.4. Statistical Analysis

Statistical analysis was conducted using SPSS^®^ v27.0 (IBM Corp., Armonk, NY, USA). Continuous variables were presented as median and interquartile ranges (IQR), while categorical variables were presented as proportions unless otherwise specified. To compare differences between groups, we used the Mann-Whitney U-test, χ2 test, or Fisher’s exact test, as appropriate. A *p*-value of 0.05 or lower was considered of statistical significance for these comparisons.

To investigate the risk factors associated with the presence of MDR/XDR bacteria, we conducted both univariate and multivariate logistic regression models using the available data. In the univariate analysis, we identified significant variables that showed associations with MDR/XDR bacteria. The multivariate analysis aimed to determine which variables remained independently associated with the occurrence of MDR/XDR bacteria after adjusting for potential confounding factors. For this second analysis, we only considered the significant variables from the univariate analysis. The results were expressed as odds ratio (OR) and 95% confidence intervals (95% CI). The multivariable logistic regression model was adjusted using the variables that had a *p*-value ≤ 0.05, which were further selected through a stepwise forward selection method (*p* in < 0.05 and *p* out < 0.10). Significant differences were indicated in bold. This approach ensures that the multivariate analysis focuses on the most influential variables, thereby providing a more accurate understanding of the relationships between the risk factors and the occurrence of MDR/XDR bacteria.

## 5. Conclusions

This study provides updated profiles of *Enterobacterales* etiology and antimicrobial susceptibility of strains isolated from different types of samples collected from patients attending a country hospital in the Eastern Region of Ghana. The main findings of this study are as follows:-The majority of samples arriving at the microbiology laboratory come from urine, the subjects being mainly women.-There is an important prevalence of AMR in all different microorganisms studied, particularly a significant number of MDR and XDR bacteria.-Males and older age groups showed a significant association with MDR and XDR bacteria.-Urine samples exhibited a significant association with XDR bacteria.

These data constitute a red flag for AMR surveillance and control in a rural area where the availability of antibiotics is affected, being expensive, and healthcare support is not always an option. Therefore, conducting microbiological studies becomes vital and justifies the implementation of culture analysis capabilities. In conclusion, regular community surveillance of antimicrobial resistance patterns is necessary to inform and recommend empirical treatments in primary care.

## Figures and Tables

**Table 1 antibiotics-12-01321-t001:** Demographic characteristics of the patients included in the study, classified by sample origin (total, urine, high vaginal swabs (HVS), blood, body fluids and aspirates (BFA), and sputum). n (number of samples in each category); N.A. (not applicable). Continuous variables were presented as median and interquartile ranges (IQR), and categorical variables as proportions. The *p*-value was obtained using the χ2 test and is considered significant when *p* ≤ 0.05.

	Total	Positive Culture (n = 872)	Negative Culture (n = 2033)	*p*-Value
SEX (n = 2905)	985 (33.9%)	Male [248 (28.4%)] Female [624 (71.6%)]	Male [737 (36.3%)] Female [1296 (63.7%)]	<0.001
Age (n = 2870)	30 (20–48)	30 (20–51)	30 (20–47)	<0.001
Leucocyte % (n = 2088)	3 (1–12)	10 (3–50)	2 (1–7)	<0.001
	Urine Samples (n = 1547)	Positive Culture (n = 352)	Negative Culture (n = 1195)	
Sex (n = 1547)	580 (37.5%)	Male [102 (29%)] Female [250 (71.0%)]	Male [478 (40%)] Female [717 (60.0%)]	<0.001
Age (n = 1525)	33 (24–56)	44 (28–63.3)	32 (23–50)	<0.001
Leucocyte % (n = 1526)	2 (1–10)	10 (2.5–50)	2 (1–5)	<0.001
	HVS (n = 570)	Positive Culture (n = 238)	Negative Culture (n = 332)	
Sex (n = 570)	N.A.	N.A.	N.A.	
Age (n = 566)	29 (24–37)	28 (23–35)	29 (25–39)	0.010
Leucocyte % (n = 562)	5 (2–15)	10 (4–20)	4 (2–12)	<0.001
	Blood (n = 439)	Positive Culture (n = 184)	Negative Culture (n = 255)	
Sex (n = 439)	237 (54.0%)	Male [106 (57.6%)] Female [78 (42.4%)]	Male [131 (51.4%)] Female [124 (48.6%)]	0.196
Age (n = 437)	1 (0.003–15)	1 (0.003–23)	0.84 (0.003–9)	0.123
Leucocyte % (N.A.)	N.A.	N.A.	N.A.	
	BFA (n = 202)	Positive Culture (n = 53)	Negative Culture (n = 149)	
Sex (n = 202)	94 (46.5%)	Male [17 (32.1%)] Female [36 (67.9%)]	Male [77 (51.7%)] Female [72 (48.3%)]	0.014
Age (n = 195)	34 (12–53)	32 (7.5–50)	35 (13–54)	0.315
Leucocyte % (N.A.)	N.A.	N.A.	N.A.	
	Sputum (n = 147)	Positive Culture (n = 45)	Negative Culture (n = 102)	
Sex (n = 147)	74 (50.3%)	Male [23 (51.1%)] Female [22 (48.9%)]	Male [51 (50.0%)] Female [51 (50.0%)]	0.901
Age (n = 147)	46 (32–62)	53 (42.5–65.5)	42 (29.8–61)	0.022
Leucocyte % (N.A.)	N.A.	N.A.	N.A.	

**Table 8 antibiotics-12-01321-t008:** Antimicrobial resistance profile of the main groups of *Enterobacterales* isolated from cultures. The main groups of antibiotics involved in Multidrug-Resistant mechanisms are shown. The number of resistant isolates is represented by the number of microorganisms tested in each category, and the percentage they imply is included. Regarding the percentage of Multidrug-Resistant bacteria (MDR), it was calculated out of the number of microorganisms tested for three or more antibiotics. The percentage of Extensive Drug-Resistant bacteria (XDR) was gauged in relation to the number of MDR bacteria. * Folate pathway inhibitors are Trimethoprim-sulphamethoxazole. ** Phenicols are Chloramphenicol.

Antimicrobial Category	*Escherichia coli*	*Klebsiella* spp.	*Proteus* spp.	*Enterobacter* spp.	Non-Identified *Enterobacterium*
Aminoglycosides	64/143 (44.8%)	58/133 (43.6%)	9/22 (40.9%)	4/11 (36.4%)	31/76 (40.8%)
Chephalosporins	95/141 (67.4%)	99/136 (72.8%)	10/22(45.5%)	9/11 (81.8%)	54/79 (68.4%)
Penicillins	44/47 (93.6%)	42/44 (95.5%)	5/7 (71.4%)	-	17/19 (89.5%)
Carbapenems	10/32 (31.3%)	5/31 (16.1%)	1/8 (12.5%)	0/3 (0%)	8/23 (34.8%)
Fluoroquinolones	75/138 (54.3%)	63/132 (47.7%)	6/23 (26.1%)	5/11 (45.5%)	39/72 (54.2%)
Tetracyclines	18/28 (64.3%)	18/24 (75.0%)	-	3/3 (100%)	10/14 (71.4%)
Nitrofurantoin	27/90 (30.0%)	38/73 (52.1%)	-	4/4 (100%)	19/40 (47.5%)
Folate pathway inhibitors *	6/6 (100%)	4/4 (100%)	1/3 (33.3%)	1/1 (100%)	3/6 (50.0%)
Phenicols **	2/5 (40.0%)	-	1/1 (100%)	0/1 (0%)	0/3 (0%)
MDR	80/140 (57.1%)	74/137 (54.0%)	7/23 (30.4%)	6/11 (54.5%)	42/78 (53.8%)
XDR	27/80 (33.8%)	30/74 (40.5%)	1/7 (14.3%)	4/6 (66.7%)	16/42 (38.1%)

## Data Availability

Data cannot be shared publicly because they are confidential. Data are available from the Department of Clinical Microbiology from Saint Dominic’s hospital in Akwatia, Ghana for researchers who meet the criteria and had been authorized to access confidential data.

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
