# Peer review of "The Antimicrobial Resistance (AMR) Rates of Enterobacterales in a Rural Hospital from the Eastern Region, Ghana: A Retrospective Study, 2022"

_antibiotics, 2023, doi:10.3390/antibiotics12081321_

Round 1

Reviewer 1 Report

In the presented manuscript, Seijas-Pereda and co-workers presented “The Antimicrobial Resistance (AMR) Rates of Enterobacterales in Easter Region, Ghana: a retrospective study, 2022”.

The manuscript is suitable for publication in Antibiotics, the language should be revised and additional analyses should be performed to present a comprehensive analysis as proposed by the authors.

I provide some major and minor comments below.

Major;

1. The authors must define the method of bacterial identification used in the study.

2. The authors refer in the tables: “Non-identified enterobacterium”, to increase the resolution of the study, the authors are recommended to identify these strains.

3. Write the conclusion highlighting in points the most important findings of the study.

Minor,

The writing and grammar of the article must be revised.

1. Line 101. A microorganism was isolated from 872 (30%) cultures. Review the sentence.

Extensive editing of English language required

Author Response

Manuscript ID: antibiotics-2540864

Title: The Antimicrobial Resistance (AMR) Rates of Enterobacterales in a rural hospital from Eastern Region, Ghana: a retrospective study, 2022.

Authors: Laura Seijas-Pereda*, Carlos Rescalvo-Casas, Marcos Hernando-Gozalo, Vida Angmorkie-Eshun, Eunice Agyei, Vivian Adu-Gyamfi, Isaac Sarsah, Maite Alfonso-Romero, Juan Cuadros-González, Juan Soliveri-de Carranza, Ramón Pérez-Tanoira*.

We would like to sincerely thank the reviewers for their comments, which have certainly enhanced the quality of the article.

REFEREE 1

Major;

  1. The authors must define the method of bacterial identification used in the study.

In the section 4. Materials and methods, subsection 4.3. Specimen’s collection and processing, we indicate how the bacteria were identified:

“Initially, samples were analyzed by microscopy using Gram staining. Subsequently, the samples were cultured on agar plates prepared in the laboratory using purified Oxoid™ Thermo Scientific™ agar. Cystine Lactose Electrolyte-Deficient (CLED) agar was used for urine samples, while Blood agar, Chocolate agar, and MacConkey agar were used for the rest of the specimens tested (HVS, blood, BFA, sputum). The bacterial isolates were characterized using colony morphology and Gram staining reaction; and through a panel of biochemical tests following the standard microbiological procedure. Gram-positive cocci were distinguished and recognized based on Gram stain, blood agar hemolysis patterns, colonial characteristics, catalase test, coagulase test, mannitol fermentation test.

Gram-negative bacteria were identified based on Gram reaction, colony morphology (visual culture characteristics of a bacterial colony on an agar plate) and pigmentation, on triple sugar iron agar (TSI) fermentation of glucose and lactose and H2S production, formation of indole, and citrate utilization, and urea hydrolysis [38].

The combination of these results typically led to the identification of a specific bacterial genus and species, as showed on Table S1. However, in some cases, certain combinations of results did not match any of the microorganisms within the scope of our technique. In such cases, we could only report the identification as far as we could determine, such as indicated in the results, "Non-identified Enterobacterium."

Also included in the discussion:

“Another significant limitation of this study is that all laboratory analyses were conducted in Ghana, where resources and availability of material are limited. Consequently, we faced numerous difficulties in identifying some bacterial isolates to genus and species level, as well as determining resistance mechanisms.”

  1. The authors refer in the tables: “Non-identified enterobacterium”, to increase the resolution of the study, the authors are recommended to identify these strains.

Bacterial isolates were characterized based on Gram reaction, colony morphology and pigmentation on agar plate, triple sugar iron agar (TSI), glucose and lactose fermentation and H2S production, indole formation, citrate utilization, and urea hydrolysis. Unfortunately, in some combinations of these test results, none of the most common microorganisms were identified. This is often due to the fact that they are less common strains whose identification requires more techniques. As indicated in the limitations, our study area is a rural Ghanaian setting with limited resources, making other types of testing impossible. Unfortunately, we can only report as far as we can identify, as "Enterobacteriaceae not identified", as indicated in the text at L360-364.

  1. Write the conclusion highlighting in points the most important findings of the study.

We have rewritten the conclusions following your suggestions to make them more clear. The final conclusions are as follows:

“This study provides updated profiles of Enterobacterales etiology and antimicrobial susceptibility of strains isolated from different types of samples collected from patients attending a country hospital in the Eastern Region of Ghana. The main findings of this study are as follows:

  • The majority of samples arriving at the microbiology lab come from urine, the subjects being mainly women.
  • There is an important prevalence of AMR in all different microorganisms studied, particularly a significant number of MDR and XDR bacteria.
  • Males and older age groups showed a significant association with MDR and XDR bacteria.
  • Urine samples exhibited a significant association with XDR bacteria.

These data constitute a red flag for AMR surveillance and control in a rural area where the availability of antibiotics is affected, being expensive, and healthcare support is not always an option. Therefore, conducting microbiological studies becomes vital and justifies the implementation of culture analysis capabilities. In conclusion, regular community surveillance of antimicrobial resistance patterns is necessary to inform and recommend empirical treatments in primary care.”

Minor,

  • The writing and grammar of the article must be revised.

 Thank you, we have revised the English language throughout our article.

  1. Line 101. A microorganism was isolated from 872 (30%) cultures. Review the sentence.

We reviewed the sentence hoping to make it more clear: “At least one microorganism was isolated from growth on the agar plate in 872 (30%) cultures”.

Reviewer 2 Report

The manuscript entitled "The Antimicrobial Resistance (AMR) Rates of Enterobacteriaceae in Easter Region, Ghana: a retrospective study, 2022" provides insight into the surveillance and development of AMR in people in Ghana. 

AMR is an important issue from One Health's perspective and may be a leading cause of worldwide mortality either directly or indirectly. Hence, surveillance and analysis of AMR development are necessary throughout the world, particularly in developing and underdeveloped regions. 

This article is focused on patient data and providing important information, however, the gearing of data and methodology to estimate the AMR is needed improvement. The followings are the comments that need to address in this manuscript for quality:

1. The overall writing needs improvement particularly sentence structuring and grammatical/spelling mistakes.

2. Table 1, need rearrangement to make data more readable and understandable. What is the significance of leucocyte % in this table?

3. What does mean by microorganisms heading in the result? It can be improved and written as identification of microbes etc. 

4. Authors only used culture characteristics to identify microbes. This is not sufficient to declare particular microbial species. Furthermore, the methodology must be elaborated for culture isolation and identification. 

5. Finally, I suggest improving the section on risk factors in tables as well as in text. 

The quality of the writing is good, however, grammatical errors need rechecking. Furthermore, I suggest rechecking the spelling and scientific names to ensure the quality of the manuscript. 

Author Response

Manuscript ID: antibiotics-2540864

Title: The Antimicrobial Resistance (AMR) Rates of Enterobacterales in a rural hospital from Eastern Region, Ghana: a retrospective study, 2022.

Authors: Laura Seijas-Pereda*, Carlos Rescalvo-Casas, Marcos Hernando-Gozalo, Vida Angmorkie-Eshun, Eunice Agyei, Vivian Adu-Gyamfi, Isaac Sarsah, Maite Alfonso-Romero, Juan Cuadros-González, Juan Soliveri-de Carranza, Ramón Pérez-Tanoira*.

We would like to sincerely thank the reviewers for their comments, which have certainly enhanced the quality of the article.

REFEREE 2

  1. The overall writing needs improvement, particularly sentence structuring and grammatical/spelling mistakes.

Thank you for your suggestion, we have revised the entire article for the English language.

  1. Table 1, need rearrangement to make data more readable and understandable. What is the significance of leucocyte % in this table?

Following your comment, we have tried to be clearer in Table 1. We have entered the information for women in table 1

On leukocyte %, we have added this sentence to the text under Table 1 (L110-111):" Leucocyte percentages were obtained by manual counting from the Gram stain of each sample, when possible, as they serve as indicators of the severity of the infection.”

  1. What does mean by microorganisms heading in the result? It can be improved and written as identification of microbes, etc.

This 2.3.6. section shows the percentages for all Enterobacterales identified in our study without separating them by sample type. ‘Microorganism’ heading was too general, so it has been replaced for ‘Enterobacterales’. We also changed the first sentence in this section to make it more concise (L205-207): ‘In this section we considered the 395 Enterobacterales from our data, without separating them for sample type. We found the most common to be E. coli (36.5%), closely followed by Klebsiella spp. (34.9%)."

  1. Authors only used culture characteristics to identify microbes. This is not sufficient to declare particular microbial species. Furthermore, the methodology must be elaborated for culture isolation and identification. 

As you have suggested, we revised this section 4.3. Specimen’s collection and processing, in order to make it more accurate and replicable.

“Initially, samples were analysed by microscopy using Gram staining. Subsequently, the samples were cultured on agar plates prepared in the laboratory using purified Oxoid™ Thermo Scientific™ agar. Cystine Lactose Electrolyte-Deficient (CLED) agar was used for urine samples, while Blood agar, Chocolate agar, and MacConkey agar were used for the rest of the specimens tested (HVS, blood, BFA, sputum).

The bacterial isolates were characterized using colony morphology and Gram staining reaction, and through a panel of biochemical tests following the standard microbiological procedure. Gram-positive cocci were distinguished and recognized based on Gram stain, blood agar haemolysis patterns, colonial characteristics, catalase test, coagulase test, mannitol fermentation test. Gram-negative bacteria were identified based on Gram reaction, colony morphology (visual culture characteristics of a bacterial colony on an agar plate) and pigmentation, on triple sugar iron agar (TSI) fermentation of glucose and lactose and H2S production, formation of indole, and citrate utilization, and urea hydrolysis [38].

The combination of these results typically led to the identification of a specific bacterial genus and species, as showed on Table S1. However, in some cases, certain combinations of results did not match any of the microorganisms within the scope of our technique. In such cases, we could only report the identification as far as we could determine, such as indicated in the results, "Non-identified Enterobacterium."

Also included in the discussion:

“Another significant limitation of this study is that all laboratory analyses were conducted in Ghana, where resources and availability of material are limited. Consequently, we faced numerous difficulties in identifying some bacterial isolates to genus and species level, as well as determining resistance mechanisms.”

Our study highlights the limitations of available resources and the importance of recognising and accurately communicating the extent of identification in the context of our study, a rural Ghana setting.

  1. Finally, I suggest improving the section on risk factors in tables as well as in text. 

Thank you for your suggestion. We explained a bit further in two parts of the paper:

Text from lines 228-239.

Univariate and multivariate analyses included age, sex, leucocytes from the Gram stain, and sample type. In the univariate analysis, we identified significant variables that showed associations with MDR/XDR bacteria which later were included in the multivariate analysis.

The univariate analysis, presented in Table 9.A, reveals significant relationships between having MDR/XDR bacteria and being male, as well as with older age. Additionally, urine samples show a significantly higher occurrence of MDR/XDR bacteria compared to other sample types, while sputum samples exhibit fewer MDR bacteria, but not XDR bacteria.

Considering XDR bacteria (Table 9.B), we found that higher age, being male, and urine samples are still associated with this occurrence. However, when doing the multivariate analysis, only being male and having urine samples are significant risk factors."

4.4. Statistical analysis

To investigate the risk factors associated with the presence of MDR/XDR bacteria, we conducted both univariate and multivariate logistic regression models using the available data. In the univariate analysis, we identified significant variables that showed associations with MDR/XDR bacteria. The multivariate analysis aimed to determine which variables remained independently associated with the occurrence of MDR/XDR bacteria after adjusting for potential confounding factors. For this second analysis, we consider only the significant variables from the univariate analysis. The results are ex-pressed as odds ratio (OR) and 95% confidence intervals (95% CI). The multivariable logistic regression model was adjusted using the variables that had a p-value ≤ 0.05, which were further selected through a stepwise forward selection method (p_in < 0.05 and p_out < 0.10). Significant differences are indicated in bold. This approach ensures that the multivariate analysis focuses on the most influential variables, thereby providing a more accurate understanding of the relationships between the risk factors and the occurrence of MDR/XDR bacteria.

Author Response

Manuscript ID: antibiotics-2540864

Title: The Antimicrobial Resistance (AMR) Rates of Enterobacterales in a rural hospital from Eastern Region, Ghana: a retrospective study, 2022.

Authors: Laura Seijas-Pereda*, Carlos Rescalvo-Casas, Marcos Hernando-Gozalo, Vida Angmorkie-Eshun, Eunice Agyei, Vivian Adu-Gyamfi, Isaac Sarsah, Maite Alfonso-Romero, Juan Cuadros-González, Juan Soliveri-de Carranza, Ramón Pérez-Tanoira*.

We would like to sincerely thank the reviewers for their comments, which have certainly enhanced the quality of the article.

REFEREE 3

  • Please describe Table 1 separately for women and men for clarity. This notation masks the data for women.

Thank you for bringing that to our attention, we understand it may cause confusion. We have entered the information for women in table 1

  • The number of samples detected does not match the number of samples listed in each table.

Thank you for your comment, we understand the confusion.

Problem lies in the last group of Tables 4-6 labelled as “Non-identified Enterobacterales”. This was a mistake because not all of them were ‘non-identified’ and the specimens included on the section were only explained in each table caption.

We have changed the name of this groups in the three tables for “Enterobacterium”. To make it clearer, we have also added a paragraph on each section text (not only at the table caption) explaining the specimens included on each group and why are they represented altogether for sensitivity analysis. For example:

2.3.3. Blood

A small number of Enterobacterales were isolated from blood cultures, totaling 9 isolates. Among them, six were non-identified Enterobacterium, two Klebsiella spp. and one Escherichia coli. Given their limited quantity, we grouped them together for sensitivity analysis.”

Once specified and taking these numbers into account, the additions do match the total numbers shown in Table 1.

  • The legends in the tables and the text have the same style and the tables are mixed with the same width as the text, making them difficult to read. All tables should be expanded to the full width of the page and legends should be listed at that width to differentiate them from the text.

Thank you for your comment. We have edited the table captions to their due style and expanded the tables and legends to the full width.

  • Please add the % to the notation of the results in Tables 2-6. Please add the number of tests to the notation of the results in Table 8. This will ensure that all results in the tables are expressed uniformly as 'number of resistant bacteria detected/number of tests (percentage of resistant bacteria detected)'.

Following your recommendation, we have included the corresponding percentages in the results notation for Tables 2-6 and the number of tests in Table 8, ensuring uniformity and facilitating easier interpretation of all the tables.

  • The hospitals where the analysis was conducted are small urban hospitals, albeit core hospitals, and it seems dangerous to treat this data as if it were representative clinical data for Ghana. It should be emphasized in the abstract, introduction (L73-) and conclusion sections respectively that the sample was obtained from a rural hospital and not from a large urban center. If possible, I would also like to see the words Rural and Countryside in the title. This point is in no way intended to undermine the value of rural data, but to clarify the nature of this Manuscript by differentiating it from urban data and to lead to further comparative research.

Thank you for your review; we have included the words ‘rural’ and ‘countryside’ in the title, abstract and emphasized them several times in the text.

  • L143, 153, 164. It should also be stated in the text that the isolated bacteria are of unidentified genus.

This request complements your second one. To enhance clarity regarding the isolates from each sample group, we have included in the text of the results a detailed description of the microorganisms found. Specifically, we also added this sentence in L123-125:

“It is important to note that, as explained in section 4, we categorized some isolates as 'Non-identified Enterobacterales' because we could not further identify their genus and species.

Finally, in section 4.3, 'Specimen’s collection and processing,' we explain in detail the reasons why non-identified Enterobacterales are found and how they are classified as a distinct group (L361-365). 

  • There is a lack of discussion on the points at which significant differences were found. There should be a word about why MDR was detected more in men, why it was detected at older ages and why it was detected in urine samples, respectively.

We edited the discussion section to add information about MDR in men, older ages, and urine samples, as you have suggested, in L296-307.

  • Minor comment: "Enterobacterium" might be easier to understand if it is used in the table.

We have followed your suggestion and written Enterobacterium in the tables to facilitate their comprehension.

Reviewer 4 Report

I read, The Antimicrobial Resistance (AMR) Rates of Enterobacteriacae in Easter Region, Ghana: a retrospective study, 2022, with interest. In this manuscript, the authors were aimed to assess the prevalence of Multidrug Resistant (MDR) and Extensively Drug Resistant (XDR) Enterobacterales in clinical samples from 2905 patients attended in Saint Dominic's Hospital, Akwatia, Ghana, during 2022. 

 I have some questions and suggestion.

1. Can you explain why this study is new or telling new things?

2. Abstract: Please give specific objective in this study

3. What is the prevalence of Multidrug Resistant (MDR) and Extensively Drug Resistant (XDR) Enterobacterales in clinical samples from Saint Dominic's Hospital, Akwatia, Ghana?

4. How were the samples cultured and what agar was used for each type of sample?

5. What method was used to determine antimicrobial susceptibility on site?

6. What percentage of microorganisms tested were MDR bacteria? And how many of these were XDR?

7. Which risk factors are associated with acquiring MDR/XDR bacteria according to the multivariate regression analysis performed in this study?

8. Table 7 - Vancomycin is a drug with a spectrum that exclusively targets gram-positive bacteria and should not be tested or reported for gram-negative bacteria.

9. Discussion: Please compare the results of this study with the results of other studies for a more in-depth discussion.

10. The limitations of this paper are not explicitly mentioned in the given information. However, some possible limitation could be:

•            The study did not investigate the mechanisms of antimicrobial resistance in the Enterobacterales strains, which could provide valuable insights into the development and spread of resistance.

Minor editing of English language required

Author Response

Manuscript ID: antibiotics-2540864

Title: The Antimicrobial Resistance (AMR) Rates of Enterobacterales in a rural hospital from Eastern Region, Ghana: a retrospective study, 2022.

Authors: Laura Seijas-Pereda*, Carlos Rescalvo-Casas, Marcos Hernando-Gozalo, Vida Angmorkie-Eshun, Eunice Agyei, Vivian Adu-Gyamfi, Isaac Sarsah, Maite Alfonso-Romero, Juan Cuadros-González, Juan Soliveri-de Carranza, Ramón Pérez-Tanoira*.

We would like to sincerely thank the reviewers for their comments, which have certainly enhanced the quality of the article.

REFEREE 4

  1. Can you explain why this study is new or telling new things?

Our research significantly advances the understanding of antibiotic resistance and its impact in rural low- and middle-income countries, with a particular focus on the countryside of Eastern Region, Ghana. It offers valuable insights into the various factors influencing multi-drug resistant (MDR) and extensively drug-resistant (XDR) bacterial infections, including the importance of sample type, sex, and age in determining effective treatment approaches. Furthermore, our study provides critical information about the prevalence of Enterobacterales in rural areas of LMICs, shedding light on the effectiveness of different antibiotics and the prevalence of resistances, including MDR and XDR bacteria. These findings fill gaps in existing knowledge and enable meaningful comparisons with other regions.

The implications of our research underline the urgency of implementing surveillance and control measures for antimicrobial resistance, making accurate diagnostics, studying the microorganisms, and conducting susceptibility testing essential components of effective antimicrobial stewardship. Overall, our study's contributions emphasize the importance of tailored interventions and strategies to combat antimicrobial resistance in rural settings of LMICs, where the burden is particularly high and effective treatment approaches are crucial.

  1. Abstract: Please give specific objective in this study

We added the aim of our study in the first lines of the abstract as you have recommended, L22-24.

  1. What is the prevalence of Multidrug Resistant (MDR) and Extensively Drug Resistant (XDR) Enterobacterales in clinical samples from Saint Dominic's Hospital, Akwatia, Ghana?

To define Multidrug Resistant (MDR) bacteria and Extensively Drug Resistant (XDR) bacteria, we considered only acquired antimicrobial resistance and not intrinsic resistance (if existing). MDR bacteria were defined as those non-susceptible to at least one agent in three or more antimicrobial categories, while XDR bacteria were those non-susceptible to at least one agent in five or more antimicrobial categories.

We have added the next sentence (L283-285):

" Among the Enterobacterales analyzed, approximately 50% displayed multidrug re-sistance (MDR), except for Proteus spp., with a 30%. From these MDR bacteria, 37.3% were XDR bacteria, alarmingly high for such a severe resistance mechanism."

  1. How were the samples cultured and what agar was used for each type of sample?

We have added the next sentence in the section 4. Materials and Methods, L346-350.

“Initially, samples were analysed by microscopy using Gram staining. Subsequently, the samples were cultured on agar plates prepared in the laboratory using purified Oxoid™ Thermo Scientific™ agar. Cystine Lactose Electrolyte-Deficient (CLED) agar was used for urine samples, while Blood agar, Chocolate agar, and MacConkey agar were used for the rest of the specimens tested (HVS, blood, BFA, sputum).”

  1. What method was used to determine antimicrobial susceptibility on site?

This is stated in section 4. Materials and Methods, L365-386:

“For antimicrobial susceptibility studies of Enterobacterales, we employed the Kirby-Bauer test or disc-diffusion method with a 0.5 McFarland inoculum in 0.9% NaCl saline on Mueller Hinton agar (Oxoid™ Thermo Scientific™). Various antibiotics discs were used, including meropenem, ceftriaxone, cefuroxime, sulfamethoxazole-trimethoprim, gentamicin, cefoxitin, cefixime, clindamycin, cloxacillin, chloramphenicol, erythromycin, nalidixic acid, pipemidic acid, amoxicillin/clavulanic acid, ampicillin, azithromycin, cefotaxime, ceftazidime, levofloxacin, nitrofurantoin, norfloxacin, ciprofloxacin, amikacin, and tetracycline. The choice of antibiotics and testing methods varied depending on the sample type, isolated pathogen, and availability of discs. 90 mm glass plates were used, with a maximum of 6 disks per plate.

Antimicrobial susceptibility for each antibiotic was determined by measuring the inhibition halo according to the Clinical and Laboratory Standards Institute (CLSI) interpretive chart for susceptibility test 2022 [39], the European Committee on Antimicrobial Susceptibility Testing (EUCAST), and/or the US Food and Drug Administration (FDA) guidelines. A bacterial isolate was considered non-susceptible to an antimicrobial agent if it tested resistant, intermediate, or non-susceptible based on these clinical breakpoints as interpretive criteria.

To define Multidrug Resistant (MDR) bacteria and Extensively Drug Resistant (XDR) bacteria, we considered only acquired antimicrobial resistance and not intrinsic resistance (if existing). MDR bacteria were defined as those non-susceptible to at least one agent in three or more antimicrobial categories, while XDR bacteria were those non-susceptible to at least one agent in five or more antimicrobial categories [19].

  1. What percentage of microorganisms tested were MDR bacteria? And how many of these were XDR?

Table 8 displays the antimicrobial resistance profile of the main groups of Enterobacterales isolated from cultures. The table highlights the key groups of antibiotics implicated in Multidrug Resistant (MDR) mechanisms, enabling us to calculate the percentage of MDR and extensively drug-resistant (XDR) bacteria. Among all the Enterobacterales analyzed in our study, approximately 50% exhibited MDR, with the exception of Proteus spp., which showed a lower MDR rate of 30%. Out of the total MDR bacteria, a concerning 37.3% were identified as XDR bacteria, indicating a high prevalence of this severe resistance mechanism.

  1. Which risk factors are associated with acquiring MDR/XDR bacteria according to the multivariate regression analysis performed in this study?

To investigate the risk factors associated with the presence of MDR/XDR bacteria, we conducted both univariate and multivariate logistic regression models using the available data. In the univariate analysis, we identified the significant variables that showed associations with MDR/XDR bacteria were: higher ages, being male and urine samples

The multivariate analysis aimed to determine which variables remained independently associated with the occurrence of MDR/XDR bacteria after adjusting for potential confounding factors. For this second analysis, we consider only the significant variables from the univariate analysis and the ones remaining were being male and older ages for MDR/XDR, and urine samples for XDR bacteria.

We discussed this results in the Discussion section L295-306.

  1. Table 7 - Vancomycin is a drug with a spectrum that exclusively targets gram-positive bacteria and should not be tested or reported for gram-negative bacteria.

It is correct, thank you for noticing. We deleted it.

  1. Discussion: Please compare the results of this study with the results of other studies for a more in-depth discussion.

We added three more studies for comparison in the discussion section. In total, we compared our study and findings with the studies from references: 1, 9, 12, 25, 26, 27, 28, 29, 30, 31, 32, 34, 35, 36, 37. If you think we should look for more, please let us know.

  1. The limitations of this paper are not explicitly mentioned in the given information. However, some possible limitation could be: “The study did not investigate the mechanisms of antimicrobial resistance in the Enterobacterales strains, which could provide valuable insights into the development and spread of resistance”.

We have included the limitations at the end if the Discussion section, L313-321. As you have suggested we have added a sentence about the inability for investigate the mechanism of antimicrobial resistances. They are as follows:

“Though it might not be obvious to the naked eye, the main drawback to this study would be, to our mind, its retrospective nature, which may have influenced the statistical analysis due to the availability of certain data being in short supply. As pointed out in the methods section, clinical and epidemiological data were collected retrospectively from patients’ medical records, and in some cases, some information was missing. Another significant limitation of this study is that all laboratory analyses were conducted in Ghana, where resources and availability of material are limited. Consequently, we faced numerous difficulties in identifying some bacterial isolates to genus and species level, as well as determining resistance mechanisms.

  • Minor editing of English language required.

We have reviewed the full article for the English language.

Round 2

Reviewer 1 Report

The authors have attended to the comments and corrections, and the manuscript is viable for publication.

Minor editing of English language required

Reviewer 2 Report

Dear Authors, 

Thank you for providing updated and improved version of the manuscript. 

The quality is improved. 

Reviewer 3 Report

All the points I raised have been appropriately corrected.